# Astaxanthin Alleviates Aflatoxin B1-Induced Oxidative Stress and Apoptosis in IPEC-J2 Cells via the Nrf2 Signaling Pathway

**DOI:** 10.3390/toxins15030232

**Published:** 2023-03-21

**Authors:** Yue Tian, Haoyu Che, Jinsheng Yang, Yongcheng Jin, Hao Yu, Chuanqi Wang, Yurong Fu, Na Li, Jing Zhang

**Affiliations:** 1Jilin Provincial Key Laboratory of Livestock and Poultry Feed and Feeding in the Northeastern Frigid Area, College of Animal Sciences, Jilin University, Changchun 130062, China; 2Jilin Academy of Agricultural Sciences, Changchun 130033, China

**Keywords:** IPEC-J2 cells, aflatoxin B1, astaxanthin, oxidative stress, apoptosis, Nrf2 signaling pathway

## Abstract

Aflatoxin B1 (AFB1), a typical fungal toxin found in feed, is highly carcinogenic. Oxidative stress is one of the main ways it exerts its toxicity; therefore, finding a suitable antioxidant is the key to reducing its toxicity. Astaxanthin (AST) is a carotenoid with strong antioxidant properties. The aim of the present research was to determine whether AST eases the AFB1-induced impairment in IPEC-J2 cells, and its specific mechanism of action. AFB1 and AST were applied to IPEC-J2 cells in different concentrations for 24 h. The AST (80 µM) significantly prevented the reduction in the IPEC-J2 cell viability that was induced by AFB1 (10 μM). The results showed that treatment with AST attenuated the AFB1-induced ROS, and cytochrome C, the Bax/Bcl2 ratio, Caspase-9, and Caspase-3, which were all activated by AFB1, were among the pro-apoptotic proteins which were diminished by AST. AST activates the Nrf2 signaling pathway and ameliorates antioxidant ability. This was further evidenced by the expression of the HO-1, NQO1, SOD2, and HSP70 genes were all upregulated. Taken together, the findings show that the impairment of oxidative stress and apoptosis, caused by the AFB1 in the IPEC-J2 cells, can be attenuated by AST triggering the Nrf2 signaling pathway.

## 1. Introduction

Aflatoxins are secondary metabolites that are produced by *Aspergillus parasiticus* or *Aspergillus flavus*, these fungi can contaminate crops during growth, transport, storage, and processing, making them susceptible to aflatoxins, which are also commonly found in edible tissues such as meat, eggs, and milk; aflatoxin B1 (AFB1) is the most poisonous and adverse of the aflatoxins [1,2]. Livestock species vary in their susceptibility to AFB1, with pigs being particularly sensitive to its effects [3]. The liver is AFB1’s primary target organ. Subchronic exposure of pigs to AFB1 (1807 µg AFB1/kg of feed) for 4 weeks resulted in clinical signs of AFB1 toxicity, such as liver fibrosis, liver dysfunction, and reduced body weight gain [4]. Additionally, AFB1 impairs the maturation of pig oocytes [5] and reduces the viability of pig alveolar macrophages [6]. The intestine is not only responsible for digesting and absorbing nutrients, but it also plays a crucial role in the body’s immune system by defending against harmful substances and toxins. Enterocytes are the first cells to be exposed to mycotoxins, and their concentration in the intestine is usually higher than in other tissues [7]. Exposure to AFB1 in low doses over time can hinder development, diminish the intestinal antioxidant ability, increase the pro-inflammatory cytokine production, and disrupt the integrity of the intestinal barrier in pigs [8]. IPEC-J2 cells and mice were used as in vitro and in vivo models, respectively, to estimate the consequences of AFB1 on intestinal function. The results showed that AFB1 disrupted the tight junction protein expression and reduced the IPEC-J2 cell viability, as well as impairing gut function in the mice, suggesting that AFB1 affects the intestinal health of animals [9]. One of the main pathways through which AFB1 causes tissue and cell damage is oxidative stress [10]. AFB1 increased ROS accumulation in rat hepatocytes and caused vacuolar degeneration and necrosis of hepatocytes, whereas AST attenuated AFB1-induced liver injury by promoting superoxide dismutase 1 (SOD1) expression and biliary cell transdifferentiation in rats [11]. 

Astaxanthin (AST) belongs to the lutein class of carotenoids [12]. Natural AST has physiological functions such as anti-oxidant, anti-inflammatory, anti-tumor, and immunity enhancement [13]. It has broad applications in healthcare products, pharmaceuticals, cosmetics, food additives, and the feed industry. Some mitochondrial defects are thought to contribute to the evolution of oxidative pressure, and these are viewed as main mediators of many diseases. Research has revealed that AST, as an antioxidant that targets mitochondria, can lessen the harm that originates from oxidative stress in the organism [14]. AST is able to reduce hypertension and vascular remodeling in hypertensive rats (SHR) by improving mitochondrial function and reducing ROS and H_2_O_2_ production [15]. AST can attenuate the decrease in mice heart rate caused by ochratoxin A and improve the concentration of SOD and glutathione (GSH) in tissues through the Keap1-Nrf2 signaling pathway [16]. AST is widely used in aquaculture and is beneficial for pigmentation in aquatic animals. AST is also beneficial in reducing oxidative damage and inflammatory responses caused by external environmental stimuli in aquatic animals, and improving these organisms’ immune responses [17,18]. AST used in poultry feeds can enhance the immunity of poultry, improve the nutritional value of the product, and improve reproductive functions [19,20]. The addition of AST to immature porcine oocytes preserved by the vitrification freezing method significantly improves the viability of vitrifying oocytes, enhances mitochondrial activity, reduces oxidative stress, and improves the developmental capacity of oocytes [21]. Weaned piglets’ feed consumption, everyday weight growth, and organ weights are unaffected by AST supplementation, and it can help to prolong the shelf life of pork fat by suppressing oxidative stress [22]. However, there are no reports on whether AST can ease the harm generated by AFB1 in the pig intestine, and the protective impact of AST in the pig intestine is less well understood.

Therefore, in this study, IPEC-J2 cells were used as the model, the aim was to elucidate the attenuating impact of AST on AFB1-induced impairment in IPEC-J2 cells and its particular mechanism of action, providing an academic basis for the application of AST as a feed additive to maintain the intestinal health of pigs. 

## 2. Results

### 2.1. Astaxanthin (AST) Attentuates the Reduction of the IPEC-J2 Cell Viability Induced by Aflatoxin B1 (AFB1)

The toxicity of AFB1 to IPEC-J2 cells and the protective effect of AST were determined by the cell viability assay. After IPEC-J2 cells were treated with AFB1 (1–80 μM), the cell viability was not substantially affected by AFB1 (1 μM), and the cell viability was significantly diminished by AFB1 (5–80 μM) (Figure 1A). The cell viability was closest to 80% at AFB1 (10 μM) and, therefore, AFB1 (10 μM) was selected for subsequent experiments. After treating the cells with different concentrations of AST (5–100 μM), it turned out that AST (5–100 μM) had no significant effect on the cell viability (Figure 1B). Subsequently, cells were co-treated with different concentrations of AST and AFB1 (10 μM). It was discovered that the cell viability increased incrementally, while AST increased compared to the AFB1 group, and the best enhancement was observed at 80 μM of AST (Figure 1C); therefore, for follow-up experiments, AFB1 (10 μM) and AST (80 μM) were jointly applied to treat the cells. 

### 2.2. AST Attenuates AFB1-Induced Elevation of LDH Levels in IPEC-J2 Cells

The degree of cell damage can be evaluated by the lactate dehydrogenase (LDH) content in the cell culture medium. After AFB1 (10 μM) and AST (80 μM) were jointly applied to the cells, as shown in Figure 2, in comparison with the CON group, the LDH content in the AFB1 group was considerably greater, and in comparison with the AFB1 + AST group, the LDH content was much lower, which implied that the addition of AST could lessen the degree of cell damage.

### 2.3. AST Reduces AFB1-Induced ROS Production in IPEC-J2 Cells 

Figure 3A,B show that that after AFB1 was applied with AST to treat cells, compared to the CON group, the accumulation of ROS was substantially raised in the AFB1 group. However, compared to the AFB1 group, the AFB1 + AST group led to a reduction in intracellular ROS levels.

### 2.4. AST Improves the Decrease of Antioxidant Capacity Induced by AFB1 in IPEC-J2 Cells

As illustrated in Figure 4, after AFB1 was applied with AST to treat cells, compared to the CON group, GSH and SOD degrees considerably decline, and MDA content substantially grow in the AFB1 group. The antioxidant enzymes just like GSH and SOD activity, which was meaningfully high up in the AFB1 + AST group that was compared to the AFB1 group, and MDA content was declining but not significantly. 

### 2.5. AST Attenuates AFB1-Induced Apoptosis in IPEC-J2 Cells

AFB1 was applied with AST to treat the cells, and the rate of apoptosis was detected through flow cytometry utilizing Annexin V and propidium iodide (PI) staining, as depicted in Figure 5A. Compared to the CON group, the AFB1 treatment significantly augmented the rate of apoptosis, while the addition of AST meaningfully reduced the AFB1-induced apoptosis. In comparison to the CON group, the pro-apoptotic proteins, such as Bax/Bcl-2 proportion, cytochrome C (Cyt-C), Caspase-9, and Caspase-3, were expressed more often due to AFB1, whereas treatment with AST reduced the expression of the above proteins (Figure 5B). 

### 2.6. AST Attenuates AFB1-Induced Cell Damage by Activating the Nuclear Factor E2-Related Factor 2 (Nrf2)-Heme Oxygenase 1 (HO-1) Signaling Pathway

One of the pathways through which AST exerts its antioxidant functional properties is the Nrf2 signaling pathway. In comparison with CON, the mRNA abundance of Nrf2, HO-1, NQO1, SOD2, and HSP70 was diminished in the AFB1 group. However, compared to the AFB1, the addition of AST increased the mRNA abundance of these genes (Figure 6A). This indicates that AFB1 blocked the Nrf2 signaling pathway, while AST successfully activated Nrf2, thus the antioxidant capacity of the cells was improved. As shown in Figure 6B, the protein levels of Nrf2 and HO-1 were consistent with the expression trends on the genes. In contrast to the CON group, the AFB1 group had substantially lower levels of HO-1 and Nrf2 proteins, but with the addition of AST, the HO-1 and Nrf2 protein levels were significantly increased.

## 3. Discussion

The first organ to be exposed to AFB1 is the intestine, which weakens the intestinal cell viability and disorders the expression of tight junction proteins, weakening the intestinal barrier [23]. In this study, we obtained the same results. It was noticed that the viability of IPEC-J2 cells gradually decreased with the increasing AFB1 concentration, with the cell viability decreasing to about 80% after 24 h of AFB1 (10 μM) treatment (Figure 1A). AST (5–100 μM) was safe for IPEC-J2 cells (Figure 1B). The results of research by Karimian A et al. were similar to the present study, with little effect on the cell viability following treatment of the non-tumoral breast cell line MCF 10 A with AST (1–100 μM) [24]. It was observed that when bovine oviductal epithelial cells were treated with sodium nitrate (1000 μM), the cell viability was greatly decreased; however, supplementation with AST (50–500 μM) gradually increased the cell viability [25]. In our study, when cells were treated with AFB1 (10 μM) and different concentrations of AST for 24 h, compared with the AFB1 group, the addition of AST (10–100 μM) stimulated a substantial increase in the cell viability, and AST (80 μM) had the best effect on enhancing the cell viability (Figure 1C). LDH release levels can be used to assess cell membrane damage, and it has been shown that LDH levels were elevated in the liver of rats under the influence of AFB1 and were reduced by curcumin intervention [26]. In the present research, as shown in Figure 2, compared to the CON, the AFB1 group augmented the LDH level considerably. Nevertheless, compared to the AFB1 group, the LDH level was also significantly diminished in the AFB1 + AST group. This indicates that cells are damaged when exposed to external stimuli, and AST may reduce the external damage to cells.

AFB1 causes mitochondrial dysfunction by binding to mitochondrial membrane proteins, generating large amounts of ROS, thus causing oxidative pressure, which is also an important pathway for its toxic effects [27]. Excess ROS leads to a redox imbalance, resulting in MDA production, and excess MDA production indicates an increased lipid peroxidation. By encouraging the production of downstream antioxidant enzymes like SOD and GSH, activation of the Nrf2 pathway boosts antioxidant capability [28]. It was shown that AFB1 treatment of microglia in the mouse spinal cord resulted in a considerable increment in H_2_O_2_ levels and MDA, and a decrement in SOD activity [29]. In the present study, in comparison with the CON, cellular ROS accumulation was significantly increased in the AFB1 group, while the addition of AST greatly decreased the ROS level (Figure 3). The MDA level surged, and the GSH and SOD activities declined in the IPEC-J2 cells due to AFB1 (Figure 4). This indicated that AFB1 led to oxidative and/or antioxidant influences on the onset of intracellular oxidative stress and that AST reversed this trend and improved the antioxidant capacity of the cells. Therefore, bioactive chemical compounds with potent antioxidant defense mechanisms might relieve oxidative stress that is brought on by AFB1 and so reduce the damage it does to the body. Ferulic acid also has a moderating impact on oxidative stress that is brought on by AFB1, which is similar to our results. It was shown that the ROS accumulation in rats’ livers that were open to AFB1 was double that in the control animals, whereas the level of ROS after co-treatment with ferulic acid was substantially reduced compared to the AFB1 group. Meanwhile, the SOD and GST activities in rat livers were significantly improved by ferulic acid [30].

Oxidative stress can impair mitochondrial dysfunction, which can activate the intrinsic pathway of apoptosis, leading to excessive apoptosis [31]. It has been confirmed that AFB1 induced an increase in apoptosis in mouse testis cells [32]. According to the study’s findings, the AFB1 group dramatically increased the rate of apoptosis in IPEC-J2 cells when compared to the CON group, and the addition of AST significantly reduced apoptosis (Figure 5A). It has been shown that AFB1 causes apoptosis by inducing oxidative stress [33,34]. The intrinsic pathway of apoptosis is that when cells are stimulated by internal or external signals, the Bcl-2 family members, Bax and Bak, oligomerize on the outer mitochondrial membrane, leading to an increase in mitochondrial membrane permeability and the release of Cyt-C, which in turn leads to the activation of Caspase-9 and Caspase-3, triggering apoptosis [35]. In this study, compared to the CON group, Bax, Caspase-3, Cyt-C, and Bax/Bcl-2 protein expressions were significantly increased, Caspase-9 protein expression tended to increase, and Bcl-2 expression was significantly reduced in the AFB1 group. Compared with the AFB1 group, after the addition of AST, Bax, Caspase-3, Caspase-9, and Bax/Bcl-2 proteins expression were significantly decreased, Bcl-2 protein expression tended to increase, and Cyt-C protein expression tended to decrease (Figure 5B), indicating that AST could reduce the AFB1-induced cell apoptosis. Similar to our findings, AFB1 induced apoptosis in neuronal cells through ROS accumulation and the upregulation of Caspase-3 and Bax mRNA levels [36]. Furthermore, AST has been found, in multiple studies, to be capable of reducing apoptosis and alleviating oxidative stress by attenuating the mitochondrial dysfunction [31]. AST reduces ROS production and inhibits Cyt-C release and Caspase-9 and Caspase-3 expression, and activates the Nrf2 pathway to protect type II alveolar epithelial cells from the threat of oxidative stress [37]. 

Nrf2 is a transcript factor that is a significant defense pathway for cells to regulate oxidative stress-induced damage and can induce the expression of detoxification enzymes and antioxidants, just like NQO1, HO-1, SOD, and CAT [38]. Many studies have demonstrated that AFB1 prevents the Nrf2 signaling. In a study of the effects of AFB1 on rat liver, AFB1 significantly decreased the mRNA expression levels of Nrf2, NQO1, HO-1, and SOD [39]. Nrf2 is a potent activator of AST, and AST exerts its powerful antioxidant properties through the Nrf2 signaling pathway [40]. Ochratoxin A is also highly toxic, and AST protects the heart, lungs, and kidneys from ochratoxin A damage in mice by triggering the Nrf2 signaling cascade [16,41,42]. In this research, after AFB1 treatment of the IPEC-J2 cells for 24 h, in contrast to the CON, the Nrf2 and HO-1 gene levels showed a decreasing trend, and the expression of NQO1, SOD2, HSP70 genes was significantly decreased. However, the AFB1 + AST group greatly increased the expression of the genes for Nrf2, HO-1, SOD2, and HSP70; moreover, NQO1 gene expression was also increased (Figure 6A), indicating that AFB1 inhibited the Nrf2 and its downstream genes, while the Nrf2 signaling cascade may be activated by the addition of AST and improve the antioxidant capacity of the IPEC-J2 cells. HSP70 is a downstream gene of the Nrf2 pathway, which has functions in cell protection, inhibition of apoptosis, and resistance to oxidative stress. Heat stress treatment of bovine mammary epithelial cells increased the expression of HSP70, which was further enhanced by pretreatment with procyanidin B2 [43]. The study verified the protein expression of Nrf2 and HO-1, which are key proteins in the Nrf2 signaling pathway. Compared to the CON, the Nrf2 and HO-1 protein expression were consistent with the expression trend at the gene level, and AFB1 significantly decreased Nrf2 and HO-1 protein expression. Nevertheless, compared to the AFB1 group, the expression of the Nrf2 and HO-1 proteins was significantly improved by AST (Figure 6B). Similar results to this research were observed in the study by Yang BB et al. They constructed an acute cerebral infarction model in which AST pretreatment increased the Nrf2 and HO-1 proteins expression, and the levels of CAT, SOD, and GPX, indicating that AST enhanced neurological function meaningfully in rats through activating Nrf2/HO-1, reducing head cell apoptosis, and boosting antioxidant capacity [44].

AST has been extensively studied in aquaculture and medicine, but less so in pig production. Although AST has a potent antioxidant activity, its poor water solubility and stability affect its utilization in vivo and studies have shown that liposome encapsulation technology and microencapsulation embedding technology can improve the bioavailability of AST [45]. For example, chitosan nanoparticles (CS-NPs) were prepared using sodium tripolyphosphate (TPP) and AST was packed into them [46], alternatively a starch-based solution has been used as a carrier for AST [47]. Comparing the protective effect of free AST and the colloidal system (ADC) obtained from DNA/chitosan-loaded AST against H_2_O_2_-induced oxidative cell damage, the results demonstrated that the ADC nanoparticle group was significantly more effective in ameliorating the cell viability, with a 10.6% higher viability than the free AST group. It also scavenged twice as much ROS as free AST, and the AST in the ADC was absorbed by intestinal endocytosis in the epithelial cells [48]. However, these delivery systems are still unstable and costly, so in-depth research is needed to select a suitable formulation technology to broaden the application of AST in production practice.

## 4. Conclusions

In conclusion, this study showed that astaxanthin (AST), at concentrations that ranged from 5–100 μM, was secure for IPEC-J2 cells, and AST treatment at concentrations, ranging from 10–100 μM, mitigated the AFB1 (10 μM)-induced reduction in the cell viability, with the most effective concentration being 80 μM. AFB1 (10 μM) caused oxidative stress by increasing ROS accumulation and MDA levels, and anti-oxidant enzymes like SOD and GSH became less active. Moreover, AFB1 (10 μM) increased the expression of pro-apoptotic proteins, such as Cyt-C, Caspase-9, Caspase-3, and the Bax/Bcl-2 ratio. However, treatment with AST (80 μM) was able to attenuate the AFB1-induced oxidative stress and apoptosis by activating the Nrf2 signaling pathway in IPEC-J2 cells, as validated by the activation of downstream antioxidant genes like HO-1, NQO1, SOD2, and HSP70. These discoveries offer a convincing rationale for the use of AST as a protective agent to promote pig intestinal health.

## 5. Materials and Methods

### 5.1. Chemicals and Reagents

AFB1 (purity ≥ 98.0%, A606874) was purchased from Sangon Biotech (Shanghai, China) and dimethyl sulfoxide (DMSO) was obtained from Sigma Aldrich (St Louis, MO, USA). Fetal bovine serum (FBS) was supplied by Gibco (Gaithersburg, MD, USA). Astaxanthin (purity ≥ 98%) was obtained from Yuanye Bio-Technology (Shanghai, China). HyClone provided the DMEM/F12 and penicillin-streptomycin (Logan, UT, USA). Dojindo Laboratories offered the Cell Counting Kit-8 (CCK-8) for utilization in this research (Kumamoto, Japan). Kits for measuring ROS, annexin V, and propidium iodide (PI) were purchased from Beyotime Biotechnology (Shanghai, China). The assay kits of lactate dehydrogenase (LDH) and superoxide dismutase (SOD) were supplied by Jiancheng (Nanjing, China). The assay kits of malondialdehyde (MDA) and the glutathione (GSH) were purchased from Solarbio (Beijing, China). The reagents for real-time qPCR included BlasTaq 2X qPCR Mater Mix (ABM, Richmond, Canada) and cDNA synthesis kit (ABM, Richmond, Canada). The antibody against Nrf2 was provided by Abcam (Cambridge, MA, USA). Antibodies against Caspase-3, Caspase-9, cytochrome C (Cyt-C), heme oxygenase 1 (HO-1), Bax, Bcl-2, and GAPDH were supplied by Proteintech (Wuhan, China).

### 5.2. Cell Culture

The IPEC-J2 cells were supplied by DSMZ (DSMZ, Braunschweig, Germany). IPEC-J2 cells were kept active in DMEM/F12 containing 10% FBS and 1% penicillin-streptomycin and incubated at 37 °C with 5% CO_2_.

### 5.3. Cell Treatment

AFB1 was resolved in DMSO. AFB1 (100 mM) was diluted in a culture medium to the precise concentration (1, 5, 10, 20, 40, and 80 μM) needed for cell treatment. The equivalent DMSO was added to the CON, and final concentration of DMSO was not greater than 0.1% (*v/v*). AST was resolved in DMSO at a quantity of 100 mM. AST (100 mM) was diluted in culture media. Concentrations of 5, 10, 20, 40, 80, and 100 μM of AST were utilized for the follow-up experiments.

### 5.4. The Cell Viability Assay

According to the instructions of the manufacturer, an examination of the cell viability was carried out. In 96-well plates, 1 × 10^4^ cells were planted per well. While the cells achieved an approximate 70% fusion, varying concentrations of AFB1 (1–80 μM) and AST (5–100 μM) treatment were applied to the cells for 24 h. CCK8 was added at 10% of the solution per well and the cells were incubated at 37 °C for 1.5 h, the absorbance value (OD) per well was then tested at 450 nm, using an enzyme marker, and the cell viability was calculated. Finally, cells were treated with AFB1 (10 μM) and AST (80 μM) for 24 h, for subsequent experiments.

### 5.5. Lactate Dehydrogenase (LDH) Assay

The LDH levels in cell cultures was determined using the LDH kit. AFB1 was applied with AST to treat cells for 24 h. The cell culture medium was gathered for LDH detection. The corresponding reagents were added in sequence according to the manufacturer’s instructions.

### 5.6. Measurement of ROS Production

To measure intracellular ROS using a DCFH-DA kit, IPEC-J2 cells were seeded in 6-well plates. While these reached about 70% fusion, cells were treated with AFB1 and AST. The cells were then stained for 25 min in the dark with DCFH-DA (10 μM). The supernatant was discarded. Fluorescence images were captured with a fluorescent microscope (Olympus, Japan) and, using the Image J (version number v1.8), fluorescence intensity was measured on these.

### 5.7. Measurement of Malondialdehyde (MDA), Glutathione (GSH), and Superoxide Dismutase (SOD) Levels

The MDA and GSH levels were measured after cells were collected in centrifuge tubes, centrifuged and the supernatant discarded. Based on the manufacturer’s specifications, the extraction solution was added and the IPEC-J2 cells were repeatedly freeze-thawed 2–3 times to break up the cells, as well as being centrifuged. Suitable reagents were successively added after the supernatant was moved to new centrifuge tubes. The absorbance was measured by means of an enzyme marker. The BCA technique was applied to calculate the protein concentration of samples.

Following the treatment of the cells with AFB1 and AST for 24 h, the supernatant was collected in centrifuge tubes and the corresponding reagents were added sequentially, based on the manufacturer’s specifications, to evaluate SOD activity.

### 5.8. Apoptosis Assay

After culturing the cells, IPEC-J2 cells culture media was obtained, the IPEC-J2 cells were softly rinsed with PBS, trypsin was added and digested in an incubator. After adding the obtained cell culture supernatant, the mixture was centrifuged. After the supernatant was disposed of, the cells were resuspended in PBS. This was centrifuged at 1000× *g* for 5 min. After disposing of the supernatant and adding 195 µL of Annexin V-FITC binding liquid, the cells were lightly resuspended. Annexin V-FITC was then added at 5 µL and mixed thoroughly. Subsequently, 10 µL of Propidium Iodide (PI) staining solution was added and gently mixed. The mixture was incubated for 10–15 min in the dark, and assayed immediately on the flow cytometry instrument.

### 5.9. RNA Isolation and Quantitative Real-Time Polymerase Chain Reaction (PCR)

The National Center for Biotechnology Information (NCBI) provided the Nrf2, HO-1, NQO1, SOD2, HSP70 and GAPDH genes, which were then sent electronically and used to create the downstream and upstream primer sequences that used the Primer 5 program for the coding sequences (CDs). The primers were verified using NCBI’s Primer Basic Local Alignment Search Tool (BLAST). Sangon Biotech (Shanghai) synthesized the primers, while GAPDH served as the internal control. Total cell RNA was isolated using TRIZOL, and making use of 1 μg of RNA, synthetizing complementary DNA (cDNA) in accordance with the manufacturer’s instructions. An eight-tube system containing 20 μL of a fluorescence quantitative reaction system, consisted of: BlasTaq 2X qPCR Mater Mix (10.0 μL), upstream primers (0.5 μL), downstream primers (0.5 μL), ddH_2_O (7 μL), and cDNA (2.0 μL). Real-time fluorescence quantitative PCR was used with the completed eight-tube system, and the following thermocycling conditions were used: enzyme activation at 95 °C for 3 min, denaturation at 95 °C for 15 s, and annealing/extension at 60 °C for 1 min. Generally, 40 cycles are used, but this number can be changed depending on the circumstances of various primers. The 2^−∆∆ct^ approach was used in order to calculate the quantitative fluorescence data. Table 1 contains a list of the primers employed for fluorescence quantification.

### 5.10. Western Blotting

After completion of the cell culture, the RIPA protein lysate containing PMSF protease inhibitor was prepared in advance and added to six-well plates. After resting on ice for 10 min, the cells were scraped with a pre-cooled cell scraper and collected in 1.5 mL centrifuge tubes, and centrifuged at 12,000× *g* for 10 min at 4 °C. The supernatant was collected. The BCA method was used to determine and calculate the sample protein concentration, with 20 µg of total protein in each loading well. The PVDF membrane was sealed with 5% skim milk powder solution prepared with 1× TBST and sealed at room temperature on a shaker for 1 h. Primary antibody working solutions for Caspase-3, Caspase-9, cytochrome C, Nrf2, HO-1, Bax, Bcl-2, and GAPDH were prepared with a 5% skim milk powder solution and the PVDF membrane was incubated with the corresponding antibodies at 4 °C overnight. The primary antibody working solution was discarded and washed 3 times with TBST. The secondary antibody working solution was prepared using 5% skim milk powder solution, and a goat anti-rabbit immunoglobulin G (IgG)-conjugated secondary antibody was incubated with the PVDF membranes for 2 h at room temperature. The secondary antibody working solution was discarded and the membrane was washed 3 times with 1× TBST. On a chemiluminescence imaging analyzer, the protein band was visualized using the Pierce Emitter Coupled Logic (ECL) substrate, and protein band density was quantified using Image J.

### 5.11. Statistical Analysis

All experiments were replicated independently no less than three times. The experimental data were statistically analyzed using SPSS (version v23.0) software, and a one-way ANOVA analysis algorithm was employed. Statistical graphs were generated using the GraphPad Prism (version v8.0) software.

## Figures and Tables

**Figure 1 toxins-15-00232-f001:**
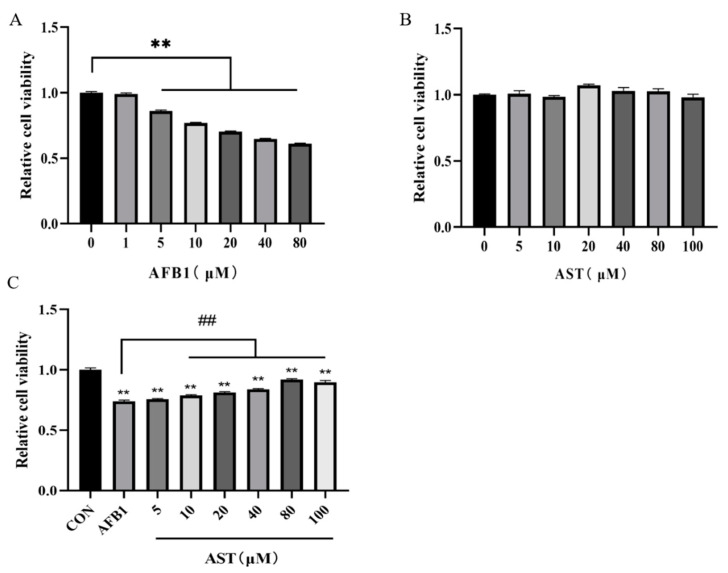
Astaxanthin (AST) attenuates the reduction of the IPEC-J2 cell viability induced by Aflatoxin B1 (AFB1). (**A**) The cell viability of AFB1-treated IPEC-J2 cells after 24 h. (**B**) The cell viability of IPEC-J2 cells after 24 h treatment with AST. (**C**) The cell viability of IPEC-J2 cells after 24 h treatment with AFB1 (10 μM) and AST (5-100 μM). The mean SEM of all values is used (*n* = 6). ** *p* < 0.01, compared with the CON group. ## *p* < 0.01, compared with the AFB1 group.

**Figure 2 toxins-15-00232-f002:**
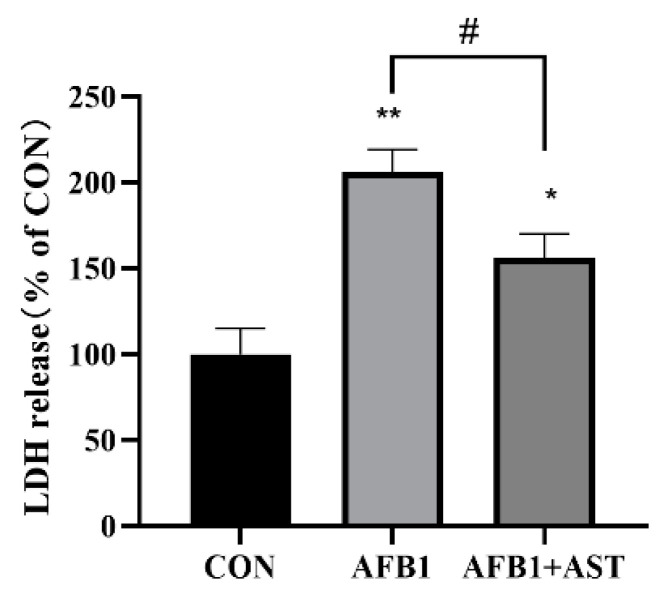
AST attenuates AFB1-induced elevation of LDH levels in IPEC-J2 cells. AFB1 (10 μM) and AST (80 μM) were jointly applied to treat the cells for 24 h. Each experiment was repeated 3 times. The mean SEM of all values is used (*n* = 3). * *p* < 0.05; ** *p* < 0.01, by comparison with the CON group. # *p* < 0.05, by comparison with the AFB1 group.

**Figure 3 toxins-15-00232-f003:**
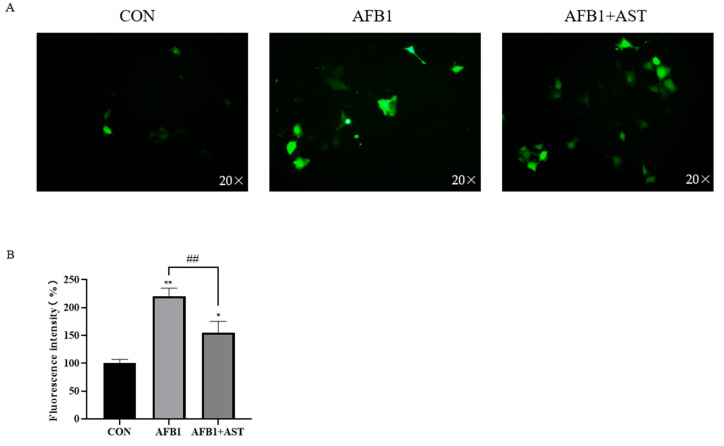
AST reduces AFB1-induced ROS production in IPEC-J2 cells. Cells were treated with AFB1 (10 μM) and AST (80 μM) for 24 h. (**A**) After incubated with DCFH-DA, cells were checked over by fluorescence microscopy AST (scale bar stands for 50 μm). (**B**) ROS fluorescence intensity in IPEC-J2 cells was analyzed making use of Image-J software (version 1.8). Each experimentation was repeated 3 times. The mean SEM of all values is used (*n* = 3). * *p* < 0.05; ** *p* < 0.01, by comparison with the CON group. ## *p* < 0.01, by comparison with the AFB1 group.

**Figure 4 toxins-15-00232-f004:**
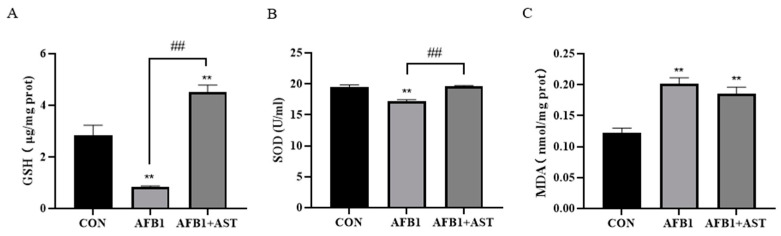
AST improves the decrease in the antioxidant capacity induced by AFB1 in IPEC-J2 cells. AFB1 (10 μM) was applied with AST (80 μM) to treat cells for 24 h. (**A**) The GSH content in IPEC-J2 cells. (**B**) SOD activity in IPEC-J2 cells. (**C**) The MDA level in IPEC-J2 cells. Each experiment was repeated three times. The mean SEM of all values is used (*n* = 3). ** *p* < 0.01, compared with the CON group. ## *p *< 0.01, compared with the AFB1 group.

**Figure 5 toxins-15-00232-f005:**
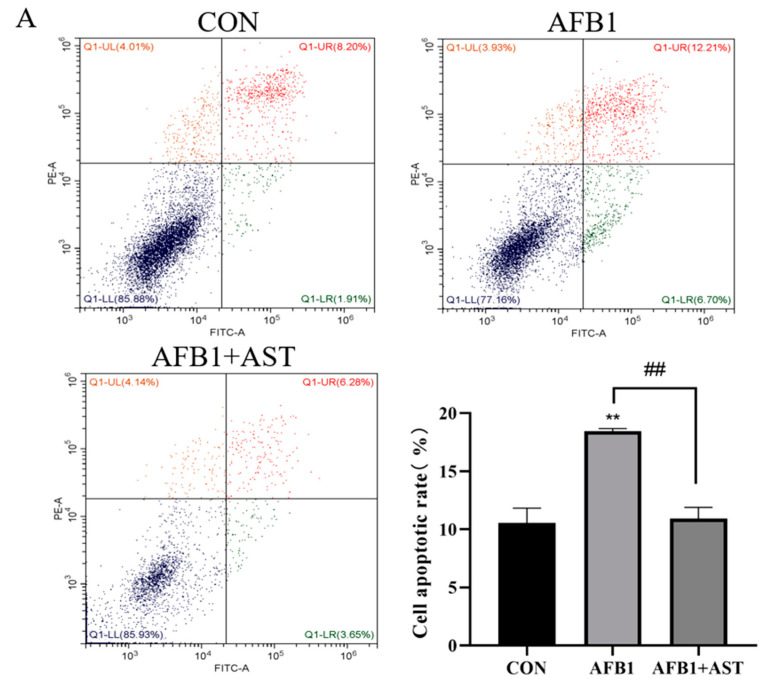
AST attenuates AFB1-induced apoptosis in IPEC-J2 cells. Cells were treated with AFB1 (10 μM) and AST (80 μM) for 24 h. (**A**) The apoptosis rate was observed by Annexin V and propidium iodide (PI) staining. (**B**) The protein levels of Bcl-2, Bax, Bax/Bcl-2, Cyt-C, Caspase-9, and Caspase-3. Each experiment was repeated three times. The mean SEM of all values is used (*n* = 3). * *p* < 0.05; ** *p* < 0.01, in comparison to the CON group. # *p* < 0.05; ## *p* < 0.01, in comparison to the AFB1 group.

**Figure 6 toxins-15-00232-f006:**
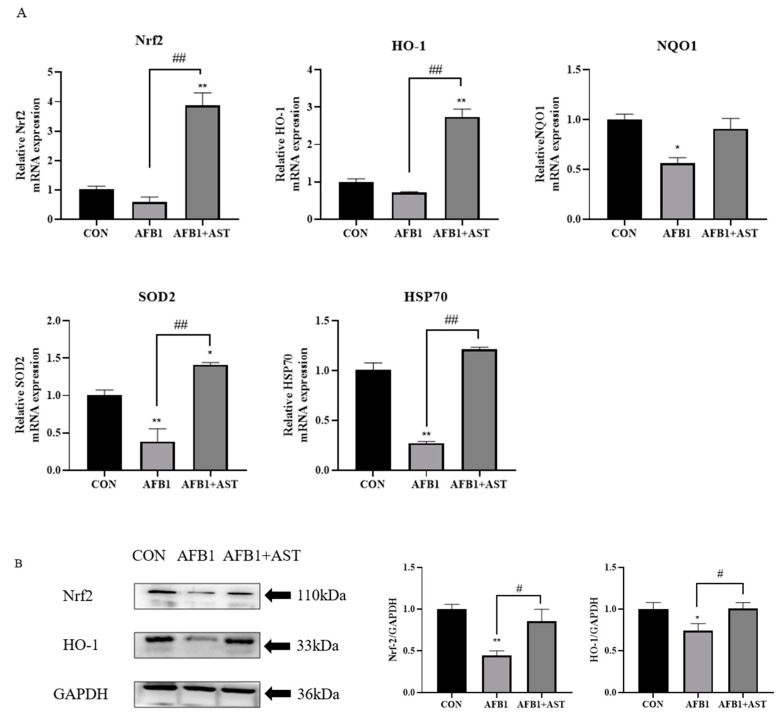
AST attenuates AFB1-induced cell damage by activating the Nuclear factor E2-related factor 2 (Nrf2)-Heme oxygenase 1 (HO-1) signaling pathway. Cells were treated with AFB1 (10 μM) and AST (80 μM) for 24 h. (**A**) The Nrf2, HO-1, NQO1, SOD2, HSP70 mRNA levels in IPEC-J2 cells. (**B**) Protein levels of Nrf2 and HO-1. Each experiment was repeated three times. The mean SEM of all values is used (*n* = 3). * *p* < 0.05; ** *p* < 0.01, compared to the CON. # *p* < 0.05; ## *p* < 0.01, compared to the AFB1 group.

**Table 1 toxins-15-00232-t001:** Gene name and PCR primer sequences.

Gene	GenBank Accession No.	Forward Primer (5′→3′)	Reverse Primer (5′→3′)	Product Size (bp)
Nrf2	XM_013984303.2	AGCACAACACATCCCGTCAGAAAC	GAGCCTGGTTAGGAGCAATGAAGAC	134
HO-1	NM_001004027.1	CCAGGTCCTCAAGAAGATTGCTCAG	GGGTCATCTCCAGAGTGTTCATTCG	146
NQO1	NM_001159613.1	GTGGAAGCCGCAGACCTTGTG	CATGGCAGCGTATGTGTAAGCAAAC	123
SOD2	NM_214127.2	TGTATCCGTCGGCGTCCAAGG	TCCTGGTTAGAACAAGCGGCAATC	93
HSP70	NM_001123127.1	CCTTACTTGGCTGGAGCACAACC	CCTGGAGAAGATGGGACGACAAATC	97
GAPDH	NM_001206359.1	GAGTGAACGGATTTGGCCGC	AAGGGGTCATTGATGGCGAC	91

## Data Availability

Data sharing is not applicable.

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
