# Peer review of "Astaxanthin Alleviates Aflatoxin B1-Induced Oxidative Stress and Apoptosis in IPEC-J2 Cells via the Nrf2 Signaling Pathway"

_toxins, 2023, doi:10.3390/toxins15030232_

Round 1
Reviewer 1 Report
The authors tested the theory that antioxidants can play as inhibitors of several aflatoxin B1 induced cell degradation processes like apoptosis. The topic is interesting and the quality of the paper is overally good.
At lane 28, I prefer omit "accidentally" as at most cases the root problem is about the feeding material.
Lane 110: on the Figure 1, AFB1 treatment is from 1-80 uM. In the text, it is 5-80uM.
Lanes 112 and 113: the AST treatment is 5-100 uM on the Figure 1, and not 0-100uM.
I can not see the explanation why 10 uM AFB1 was applied with 80 uM AST.
For Section 2.3: it is possible that you detected self-fluorescence of AFB1?
Section 5.6: the method is not described. Who is the manufacturer? What is the basis of the method?
Reviewer 2 Report
The manuscript "Astaxanthin Alleviates Aflatoxin B1-Induced Oxidative Stress and Apoptosis in IPEC-J2 Cells via the Nrf2 Signaling Pathway" describes interesting aspects for the detoxification of alflatoxin B1 (AFB1). Astaxantin (AST) is a substance providing excellent antioxidant properties.
The overall scientific work per se, especially the experimental design including the control of the putative pathways for the detoxification is of value for toxicology.
The results are presented in a set of 6 figures. The flowcytometric scatter plot (Fig. 5 A) needs a better description of the method. In the present version of Materials and Methods there is no indication how the cells were removed from the cell cultures. Further, the supplier of the annexin is not provided.
The main criticism is the unclear text in Introduction and Discussion - these passages are very long and give no stratified guide to the pathophysiological basis of this experimental work. In the introduction the species of interest - pig - is mentioned quite late - and it has to be explained why pigs are in the focus of toxicology. Further, the reader has to be informed about the specific contribution of the IPEC-J2 cell line to the research in toxicology. Please carefully check each sentence whether it is necessary. E.g. in lines 26/27 "Mycotoxins is [correct: are] widely distributed, in the growth, transportation, storage and processing of crops and other aspects are vulnerable, there are vulnerable…" What is the meaning of this phrase? At the end of the Introduction the reader must understand why the study was performed and what are the key tools (in terms of e.g. cell culture system). The reduction of the Introduction by one third is necessary.
In the Discussion there are again unclear phrases: A sentence like "…As an important part of the digestive system, the intestine is the main place for nutrient absorption … " is dispensable - are there other places for nutrient absorption? lines 206/207. And, many single experimental findings are enumerated - but as independent facts not integrated in the discussion of the present results of the study. The difficulty of AST as potential compound to reduce the toxicity of AFB1 is limited by its water solubility. The authors circumvented this problem in their experiments using DMSO. But that technique is far from an option in the use of AST in pig production.
In summary, a manuscript with a sound scientific base - however the writing is below the quality of the scientific value.
Reviewer 3 Report
The subject described in the manuscript has been explored before (e.g., see doi: 10.1292/jvms.18-0690), so the novelty of the study should be highlighted. The experimental approaches used seem correct and the data reported are acceptable. Although the manuscript is well organized and written, the following revision items should be addressed:
- L.59-60: Why the aim of the study is stated here in the middle of the Introduction section?
- L.70-94: Those lines should be greatly summarized, to focus only on the relevant background subjects related to the study.
- L.350: At which level(s) AST could atenuate the toxic effects of AFB1? Please elaborate on the Conclusion section.
Reviewer 4 Report
The manuscript fits into the scope of the journal and it seems to be well designed and executed. The results will provide and contribute to better understanding the potential of AST in porcine farming. It adds new knowledge about the benefits of this antioxidant that has been extensively studied for aquaculture.
Figures and imagines contribute to clarify the method and the results.
The wholes manuscript is well written.
Altough the actual version of the manuscript shows a good quality there are some minor recommendations to improve its quality.
- The trend of including the materials and methods at the end it is less confortable than before the results. Does this respond to the authors' guidelines or could it be changed?
- The conclusions section is too short and should be completely rewritten.
- As said before about the conclusions, the statistical study is poorly described.
